# On the Reliability of Wireless Sensor Networks with Multiple Sinks

**DOI:** 10.3390/s24175468

**Published:** 2024-08-23

**Authors:** Vladimir Shakhov, Denis Migov

**Affiliations:** 1Department of Electrical and Computer Engineering, University of Ulsan, Ulsan 44610, Republic of Korea; 2Institute of Computational Mathematics and Mathematical Geophysics, 630090 Novosibirsk, Russia; mdinka@rav.sscc.ru

**Keywords:** wireless sensor networks, network convergence, multiple sinks, system reliability, random graph, heuristic algorithm

## Abstract

The convergence of heterogeneous wireless sensor networks provides many benefits, including increased coverage, flexible load balancing capabilities, more efficient use of network resources, and the provision of additional data by different types of sensors, thus leading to improved customer service based on more complete information. However, despite these advances, the challenge of ensuring reliability and survivability remains due to low-cost sensor requirements and the inherent unreliability of the wireless environment. Integrating different sensor networks and unifying protocols naturally leads to the creation of a network with multiple sinks, necessitating the exploration of new approaches to rational reliability assurance. The failure of some sensors does not necessarily lead to a shutdown of the network, since other sensors can duplicate information and deliver data to sinks via an increased number of alternative routes. In this paper, the reliability indicator is defined as the probability that sinks can collect data from a given number of sensors. In this context, a dedicated reliability metric is introduced and examined for its effectiveness. This metric is computed using an algorithm rooted in the modified factoring method. Furthermore, we introduce a heuristic algorithm designed for optimal sink placement in wireless sensor networks to achieve the highest level of network reliability.

## 1. Introduction

Wireless sensor networks (WSNs) have gained a lot of attention due to their many applications in various fields of human activity. WSNs provide the foundational infrastructure for Internet of Things (IoT) applications [1] and exploit their ability to provide real-time monitoring and data collection through a network of spatially distributed sensors, which can measure environmental parameters such as temperature, pressure, speed, humidity, and illumination. These sensors communicate wirelessly and use multi-hop communication to relay data to a sink node, which acts as a central point for data aggregation and processing or serves as a gateway to transmit data to base stations for further treatment. WSNs are noted for their simplicity of deployment, scalability, and flexibility, making them suitable for a wide range of smart environments, including smart cities, precision agriculture, healthcare, and industrial automation systems.

One of the main challenges of wireless sensor networks (WSNs) is in ensuring reliability, especially given the limited resources of sensor nodes [2]. The reliability in WSNs is critical for accurate and consistent functioning across various applications and encompasses several dimensions, including robust data transmission, resilient network nodes, and consistent performance despite channel interferences and node failures. However, the application of reliability mechanisms typical of traditional networks is challenging in WSNs. The very limited battery capacity of the sensor nodes leads to rapid energy depletion, causing sensors to shut down and resulting in a loss of network connectivity [3]. Furthermore, the modest computing capabilities and limited storage capacity of the sensor nodes can lead to their temporary exclusion from network collaboration. At the same time, the fact that sensors are relatively inexpensive allows their redundancy to be provided in the maintenance of network operability. The self-organizing and self-healing capabilities of the WSN nodes ensure the dynamic restructuring of the network topology in response to multiple sensor failures, allowing the network to continue normal operation. Therefore, in order to find a compromise between the cost of the system and its reliability, tools are required to assess the reliability of the network under sensor failure conditions.

The convergence of heterogeneous WSNs facilitates the seamless integration of diverse sensor types and communication protocols, promoting more effective solutions for various vital IoT systems such as smart city and environmental monitoring through enhanced data accuracy, broader coverage, and adaptable user-driven applications [4]. The convergence of different WSNs should be noted as naturally resulting in a situation with multiple sinks. This circumstance can potentially have a beneficial effect on the reliability of the converged network. Multiple sinks can create redundant paths for data transmission between the sensors and the sinks. If one path becomes unavailable due to intermediate node failure or other factors, the sensors can route their data through alternate paths to reach the sinks. Therefore, it is possible to ensure reliable data collection even if several sensors have failed. Furthermore, the distribution of the data transmission load among multiple sinks serves to equilibrate the network traffic and alleviate congestion at individual sinks. Consequently, this diminishes the likelihood of packet loss attributable to a sink buffer overflow while concurrently reducing the average delay experienced by packets residing within the sink buffer.

While employing multiple sinks is evidently beneficial for enhancing the reliability of a network, it is imperative to undertake concerted efforts aimed at fortifying the overall reliability of these networks. In this paper, we consider approaches for assessing the reliability of WSNs with multiple sinks. In fact, the paradigm of WSNs requires a shift from the traditional end-to-end reliability concept to a collective event-to-sink reliability concept. This opinion posits that the WSNs can maintain effective operation even if some sensors fail, provided that a sufficient number of functional sensors remain to perform monitoring tasks. Consequently, we define WSN reliability as the probability that the number of sensors capable of delivering collected data to any sink meets or exceeds a specified threshold. As long as the number of failed sensors remains below a certain threshold, the system continues to function normally, and the remaining sensors provide sufficient data coverage and maintain the information quality required to achieve monitoring goals. This redundancy allows WSNs to reliably support various applications such as structural health diagnostics, flood warning, fire detection, Internet of Vehicles traffic management, etc. Furthermore, the requirement for sink connectivity through operational sensors may either be accepted or disregarded. The lack of necessity for connectivity between sinks characterizes the converged WSNs, wherein sinks can independently establish connections to the decision center via a backbone network. For both scenarios, with and without the requirement for sink connectivity, we propose algorithms for the calculation of requirement. Based on these algorithms, we developed a heuristic algorithm for sink placement aimed at enhancing the reliability of WSNs.

The rest of this paper is organized as follows: In the next section, we describe the related research. Section 3 introduces the main notations and system model. The algorithms are presented in Section 4. In Section 5, sink placement optimization is investigated, and the conclusions are presented in Section 6.

## 2. Related Work

Significant research has been conducted on network reliability and connectivity analysis [5], with a particular emphasis on WSNs [6,7]. Approaches utilizing random graphs to model networks have been widely used for the analysis of reliability [8,9,10]. A network is represented by a graph whose vertices correspond to network nodes, and the edges correspond to communication links. The assumption for the elements of the graph (vertices and/or edges) is that the probabilities of their presence are provided, which correspond to the reliability (availability, survivability, etc.) of network nodes and channels. Network reliability is defined as some function defined on a random graph. A frequently used measure of network reliability is the connectedness probability of a random graph, subject to independent edge failures with identical probability and absolutely reliable nodes [11]. This metric is known as all-terminal reliability.

The authors of [12] analyzed the all-terminal reliability of wireless networks, including random mesh networks. Each node was connected to neighbors via redundant radio-broadcast modules, and if a module failed, all connected links were cut off. To improve reliability, a fault-tolerant method using redundant radio modules was proposed. The link cut-offs are assumed to not be caused by channel fading after initial installation and that the number of links of each node was bounded. An estimate of the network’s cost-effectiveness was also presented. In [13], the all-terminal reliability measure was utilized to compare common commercial architectures, such as control signal transport in aircraft, under low- and high-stress conditions. The analysis demonstrated that Harary graphs exhibit favorable reliability properties. Additionally, it was demonstrated that the network design for independent link failures should prioritize reliability under high stress, as low-stress reliability was less affected by graph structure; achieving moderate reliability performance under high stress necessitates very high node degrees and small network diameters. The authors of [14] emphasized the importance of evaluating the reliability of multi-autonomous underwater vehicle cooperative systems, identifying all-terminal reliability as a key index while considering factors such as system topology and underwater acoustic communication.

The application of graph-based models with random edges, as well as the concept of terminal reliability, extends beyond the realm of wireless networks, in which the links are inherently unreliable. The following examples include optical networks. In [15], the reliability of generalized multiprotocol label switching-controlled optical networks was considered, examining reductions in long-distance communication costs. The authors of [16] addressed the challenge of efficiently designing a disaster-resilient wireless-link-augmented optical network infrastructure. An optimization model was formulated to identify the subset of links in an optical network topology whose wireless augmentation maximized the post-disaster recovery of overall network availability within a given budget constraint. In [17], the authors investigated the reliability of a four-fiber bidirectional line-switched ring.

There are different interpretations of connectivity in the context of network reliability. In some applications, it is often sufficient to ensure connectivity for a subset of the graph nodes (k-terminal reliability, two-terminal reliability) [7,13,17,18,19]. In some cases, it is reasonable to take a diameter constraint into account [20]. The reliability indices also include average pairwise connectivity [21] and the average size of a connected subgraph that contains the central node [22].

The case of unreliable nodes is also presented in the literature, which also introduces diverse interpretations of network reliability. One such criterion [23] posits that a network maintains functionality if a designated set of nodes remains interconnected. Alternatively, other perspectives have defined network operability based on the interconnection of all currently functional nodes. This is a concept known as residual connectivity reliability and is discussed in [24,25]. However, these approaches overlook the distinctive characteristics inherent to WSNs. Neither of these reliability metrics adequately capture the operational dynamics of WSNs; too few nodes might persist within the network to sustain effective monitoring, yet the network would still be deemed reliable according to these criteria. Additional efforts are essential to adapt network reliability indicators specifically for WSNs.

In [26], the authors considered WSN reliability as the probability that there is an operational communication path between the sink node and at least one operational sensor in a target cluster. The expected length of the shortest of such paths was also investigated. The authors proposed methods for the exact calculation of these WSN characteristics and studied some exceptional cases which could be solved in polynomial time. All the results were obtained assuming that a network contains only one sink and that the sensors were imperfect while the links were perfectly reliable. The authors of [27] developed auxiliary models of network element availability based on the theory of Markov processes to take into account the functional features of WSNs (the mobility of the sensor, energy consumption, etc.). The technique presented in the paper enables the application of structural reliability results to the calculation of functional reliability indicators, while incorporating the features of network topology. In [28], a new reliability index of the WSNs was defined as the probability that sinks collect data from a given number of sensors; however, the option of the absence or presence of the requirement of sink connectivity was not implemented, special cases of network topologies were not considered, and the problem of the optimal placement of sinks was not investigated. To the best of our knowledge, the problem statement presented in this paper is both comprehensive and original; the algorithms we present have no known counterparts.

## 3. System Models

Let us represent the WSNs topology with an undirected graph G=(V,E,S), where *V* is a set of network nodes (sensors and sinks), *E* is a set of edges, and S is a set of sinks, S⊂V. Let *N* be the number of sensor nodes (i.e., $N=\left|V\right|-|S|)$. We assume that the set S contains at least one element. The network nodes are subject to random failures with given probabilities. Furthermore, we refer to the probability of a node’s existence as its reliability. We denote the reliability of *v* node as *p_v_*. The reliability on links is assumed to be much higher than the reliability of the sensors. If the distance between two sensors does not exceed the transmission range of the sensors, an edge is created between the corresponding vertices in the graph; otherwise, no edge is formed. At the short distances typical in WSNs, Automatic Repeat Request (ARQ) or Forward Error Correction (FEC) mechanisms can ensure near-perfect channel reliability. Therefore, the edges in the graph G are assumed to be perfectly reliable, that is, if an edge is present, then its reliability is equal to 1. Without reducing generality, the sinks are also considered to be perfectly reliable. Otherwise, we have to perform a preliminary step algorithm of reliability calculation, which is described below.

The WSNs are assumed to be able to effectively operate even if some sensors fail, but there are still enough workable sensors to perform monitoring. The number of workable sensors should not be less than a given threshold, which we denote by *T*. Since the sinks are assumed to be perfectly reliable, we arrive at the following restriction on *T*: 1 ≤ *T* ≤ *N*.

Another requirement for the effective operation of WSNs is the sink connectivity. Let us introduce their definitions. An elementary event is a special realization of the graph defined by the existence or the absence of each node. The existing nodes will be referred to as operational, and the absent ones will be referred to as faulty. The variable $V_{Q}$ denotes a set of all existing nodes. The probability of an elementary event is equal to the product of probabilities of the existence of operational nodes multiplied by the product of probabilities of the absence of faulty nodes, as follows:(1)$$p\left(Q\right)=\prod_{v\in V_{Q}}p_{v}\prod_{v\notin V_{Q}}\left(1-p_{v}\right)$$

We define a random variable Y using the following rule: Y(Q) is the number of nodes from $V_{Q}/S$ that are connected to any sink. An elementary event Q is called successful if all sink nodes are connected with each other by nodes from $V_{Q}$ and Y(Q)≥T. Otherwise, it is called unsuccessful. The reliability of the WSNs is defined as the probability of the event consisting of all successful events and of only those events, which is denoted as $R_{S,T}(G)$.

However, the condition of the sink connectivity is not suitable for all types of WSNs, as has already been mentioned above. Therefore, we also consider another reliability measure for describing such WSNs. In this case, an elementary event Q is called successful if Y(Q)≥T. The probability of the event, consisting of all successful events in this meaning, is denoted as $R_{\bar{S},T}(G)$.

To summarize, we have defined two reliability measures, $R_{S,T}(G)$ and $R_{\bar{S},T}(G)$. $R_{S,T}(G)$ is the probability of the intersection of the following two events: sinks are connected with each other, and at least *T* workable sensors are connected to those sinks. $R_{\bar{S},T}(G)$ is the probability of only the second event. Below, we propose methods for their calculation.

## 4. Modification of the Factoring Method

It is convenient to perform the calculation of $R_{S,T}\left(G\right)$ and $R_{\bar{S},T}(G)$ using the well-known factoring method [29], which should be modified for this purpose. The factoring method is a recursive procedure which is based on the total probability law. The method decomposes the probability space into two subspaces according to two hypotheses on the success or failure of the particular network element. The chosen element is called a pivot element. As a result, we obtained two new graphs. In one of them, the pivot element is absolutely reliable with a reliability value of 1 (a branch of contraction), and the second one is absolutely unreliable with a reliability value of 0 (a branch of removal). The probability of the first event is equal to the reliability of the pivot element; the probability of the second event is equal to the failure probability of the pivot element. Further, the obtained graphs are subject to the same procedure. We reduce the number of random elements incrementally. The total probability law provides an expression for the network reliability. Below, we provide the expression for the reliability of the *WSN* using the *e* element as the pivot:(2)$$R\left(WSN\right)=p_{e}R\left(WSN|e^{+}\right)+{(1-p}_{e})R\left(WSN|e^{-}\right)$$
where *R* is the chosen reliability measure; $R\left(WSN|e^{+}\right)$ is the reliability of a network when the element *e* is workable; $R\left(WSN|e^{-}\right)$ is the reliability of a network when the element *e* is faulty; and $p_{e}$ is the reliability of the pivot element *e*.

The recursions continue until a graph is obtained for which the reliability can be directly calculated. In the most general case, it is an absolutely reliable graph (with 1 reliable value), or an unreliable graph (with 0 reliable value). Nevertheless, for some reliability measures, it is possible to finish the recursions earlier. For example, in the course of the k-terminal reliability calculation, the recursion can be finished if we obtain a small-dimensional graph or a graph of a special type. For the further calculation of such graph reliabilities, special formulas can be used [8].

The calculation of $R_{S,T}\left(G\right)$ is performed in the same way, but with allowance for the need for the following two conditions: the sink connectivity and the availability of a sufficient number of sensors attached to them. Thus, there are additional variants of the final graphs, i.e., the graphs for which this is the last recursion call. We offer to choose one of the sensors as the pivot element, which is adjacent to any sink, or one of the sensors adjacent to any already-passed sensor with a reliability of 1. If there is no such unpassed sensor, the reliability of the current graph is 0. In performing this selection, we accumulate the number of sensors which are connected to sinks. Similar to the original factoring method, we use the names “branch of contraction” and “branch of removal”, despite the fact that we actually do not perform either contraction or removal in a graph. We present their description below.

In the case of a branch of contraction, the number of sensors attached to the sinks is increasing. If it reaches *T*, we need to check the connectivity of sinks via absolutely reliable sensors. We denote the function for such verification by *SinksRelConnectivity*. If the check is successful, then the final subgraph corresponding to the successful event is obtained. If the check is unsuccessful, then the factoring procedure continues until we reach the sink connectivity. Therefore, we calculate the probability of sink connectivity in a graph with unreliable nodes and we denote the corresponding computing function by $R_{S}$. The algorithm for this function is also based on the factoring procedure and is presented in [23].

In the case of a branch of removal, we decrease the number of sensors which in the process of further factoring could potentially be absolutely reliable. The event can be unsuccessful due to the two reasons. The first reason is the sinks’ disconnection, and the second reason is the inability to reach the necessary number of sensors attached to the sinks. *SinksConnectivity* denotes the function for the verification of whether the sinks are connected via non-zero reliability sensors. If the check is unsuccessful, the reliability of the current graph is 0. Otherwise, we should then consider the second condition: the number of remaining sensors should be sufficient to reach *T*. Once we reach the situation where the number of remaining sensors is the minimum possible to satisfy this condition, we will declare them all absolutely reliable with the corresponding calculations. Therefore, we also obtain successful events in the branch of removal.

However, it is possible that some of these remaining sensors are not connected with any sink. For this purpose, we use a function *RelNodesConnectivity*, which checks the connectivity of the absolutely reliable, already-passed nodes (all sinks are initially marked as being passed). If checking is unsuccessful, the reliability of a current graph is 0. Otherwise, for obtaining the reliability value of a current graph, we should multiply the reliabilities of the remaining nodes before making them absolutely reliable.

Summing up, we have obtained the following expression for the factoring procedure for the RS,TG calculation:(3)RS,TG=pvRSGRS,TGv*+(1−pe)RS,TG\v
where pv is the *v* sensor reliability, Gv* is the G graph with the absolutely reliable sensor *v*, and G\v is the graph obtained by deleting *v* from G.

For the calculation of $R_{\bar{S},T}(G)$, we may contract all sinks into one sink at a preliminary step since there is no necessity for the sink’s connectivity. After that, the calculation may be performed by the method presented. However, this leads to some excess operators. Thus, we describe the factoring procedure for $R_{\bar{S},T}(G)$ calculation below. Nevertheless, the expression (3) of the factoring procedure for $R_{S,T}\left(G\right)$ is the same as for $R_{\bar{S},T}(G)$.

As in the previous case, it is convenient to choose one of the sensors that is adjacent to any sink as a pivot element or one of the sensors that is adjacent to any already-passed sensor with a reliability of 1. If there is no unpassed sensor, the reliability of the current graph is 0. In performing this selection, we accumulate the number of sensors connected to sinks. If, in the branch of contraction, the total number of sensors connected to at least one sink reaches *T*, then we obtain the final graph. In the branch of removal, we may be unable to reach the necessary number of sensors attached to the sink. If the number of remaining sensors is just sufficient to ensure this condition, we render them all absolutely reliable. However, as for the $R_{S,T}\left(G\right)$ calculation, it is possible that some of these remaining sensors are not connected with any sink. Therefore, we need to use the function *RelNodesConnectivity* again, which checks the connectivity of the absolutely reliable already-passed nodes (all sinks are initially marked as passed). If checking is unsuccessful, the reliability of the current graph is 0. Otherwise, for obtaining the reliability value of a current graph, we should multiply the reliabilities of the remaining nodes before making them absolutely reliable.

## 5. Algorithm

With each graph that arises in the factoring processes, we associate an array of node existence probabilities denoted by *P*, where *P*[i] is the probability that the node *i* is operational. The length of *P* is equal to *N*. The array $P_{0}$ of sensor reliabilities corresponds to the initial graph *G*. In using X(P), we notate a set of sensors that have already been passed by the algorithm. To check the connectivity, it is convenient to use a breadth-first search (BFS). In general, the computational complexity of BFS is $O\left(\left|V\right|+|E|\right)$. In this particular case, $\left|V\right|$ is less than or equal to $\left|E\right|+1$; therefore, the complexity can be defined as $O\left(E|\right)$. In the presented Algorithm 1, the sinks are considered to be absolutely reliable. Otherwise, the same algorithm is used, with the only difference being that the resulting value of the WSN reliability must be multiplied by the product of sink operability probabilities.
**Algorithm 1.** The algorithm for calculating WSN reliability with multiple sinks1: function *FactoringWSN*(*P*)2: **if** *∃ v ∈ V/X(P) : ∃u ∈ X(P) : e = (u*, *v) ∈ E*, *P[u] =* 1 **then**3:   *X(P) ← X(P) ∪ v*4:   *p ← P[v]*“Branch of contraction”5:   *P[v] ←* 16:   *x ← x* + 17:   *y ← y* + 18:   **if** *y* = *T* **then**9:     **if** *SinksRelConnectivity(P)* **then**$“\mathrm{Skip}\;\mathrm{for}\;R_{\bar{S},T}(G)$”      *RContract* ← *p* 10:     $\mathrm{else}\;\mathrm{RContract}\;\leftarrow \;p*R_{S}\left(P\right)$11:     **end if**12:   **else** *RContract* ← *p * FactoringWSN(P)*13:   **end if**14:   *P[v]* ← 0“Branch of removal”15:   *y ← y* − 116:   **if** *SinksConnectivity(P)* **then**$“\mathrm{Skip}\;\mathrm{for}\;R_{\bar{S},T}(G)$”17:     **if** *z− x + y* = *T* **then**18:       *r* ← 119:       **for all** *i : P[i]* > 0 **do**20:         *r ← r * P[i]*21:         *P[i] ← 1*22:       **end for**23:       **if** *RelNodesConnectivity(P)* **then**24:         *RRemoval ← (1 − p) * r*25:       **else** *RRemoval* ← 026:       **end if**27:     **else** *RRemoval* ← *(1 − p) * FactoringWSN(P)*28:     **end if**29:   **else** *RRemoval* ← 0$“\mathrm{Skip}\;\mathrm{for}\;R_{\bar{S},T}(G)$”30:   **end if**31: **end if**32: **return** *RContract + RRemoval*“Final Result”

The recursive function named *FactoringWSN* is the modified factoring algorithm for calculating the reliability $R_{S,T}(G)$ or $R_{\bar{S},T}(G)$. The initial parameter of this function is $P_{0}$, that is, the call to the function *FactoringWSN*($P_{0}$) returns the reliability value $R_{S,T}(G)$. We present Algorithm 1 for calculating the reliability of a WSN with multiple sinks in the form of pseudocode, taking into account the presence and absence of sinks connectivity requirement. In this algorithm, the variable *x* stores the number of nodes already processed, the variable *y* stores the number of nodes reliably connected to any sink, the variable *z* is initialized as *N*, and the remaining local auxiliary variables of this function are initialized to zero unless otherwise specified.

Although the presented algorithm still requires significant computational effort, it is capable of handling medium-scale WSNs (up to 100 sensors). The scaling of the introduced WSN reliability measure to large networks is achievable through approximate techniques, such as Monte Carlo methods, simulated annealing technique, bionic algorithms. However, the proposed algorithm is effective even for large networks in special cases, such as series-parallel topologies.

## 6. Heuristic Algorithm for the Optimal Sink Placement

To address the critical task of ensuring the reliability of WSNs, it is advisable to employ comprehensive strategies beyond merely maintaining additional independent routes. We also suggest enhancing WSN reliability through the optimal positioning of the sinks. This leads us to the following statement of the optimization problem. For a given WSN with the topology represented by the graph *G* = (*V*,*E*), and for given integers *T*, we need to find S *⊆ V*, which maximizes the value of $R_{S,T}(G)$.

We propose solving this problem using a well-known heuristic method initially introduced in [30] for the traveling salesman problem. Subsequently, this approach has been applied to a variety of other problems. We introduce Algorithm 2, a refined adaptation of this heuristic method specifically designed to address the specified sink placement problem. The proposed method necessitates an initial solution, which can either be generated randomly or through various approaches such as leveraging domain-specific knowledge or utilizing historical data.
**Algorithm 2.** The algorithm for multiple sink placement in WSN1: function SinksPlacement(G(V,E,S*)*, *T*)2:   R←RS,T(G)“Initial Sinks Placement”3:   *Replacement ← true*4:   **while** *Replacement = true* **do** 5:     S← V\S6:     *Replacement ← false*7:     **for all** *v ∈ S* **do**
8:       *NodeForReplacement ←* 09:       **for all** *u ∈*S **do** 10:         R*←Rv⋃S\u,T(G)11:         if R*>R **then**12:           R←R*13:           *NodeForReplacement ← u*14:         **end if**15:       **end for**16:       **if**
*NodeForReplacement ≠* 0 **then** 17:         S ← v⋃S\u18:         *Replacement ← true*19:       **end if**20:     **end for**21:   **end while**22:  return S“Obtained Sinks Placement”

## 7. Numerical Analysis

To begin with, we will illustrate the complexity of assessing the network reliability, considering topological features, through easily verifiable examples. Assessing the reliability of a network, while also considering its topological features, presents a multifaceted challenge that necessitates a comprehensive analysis of various specific circumstances, making generalizations difficult. The reliability criteria and network parameters, such as the number of sensors or the number of receivers, can have significantly different effects on the reliability.

For a star topology with a single sink in the center, the reliability indices are as follows:(4)RS,TG=RS¯,T(G)=∑i=TNNipi(1−p)N−i   

The result is exactly the same in the case of multiple sinks and a topology described by a complete bipartite graph, where one of the independent sets contains all the sinks; a quasi-bipartite graph, where the sinks also form an independent set, and the remaining vertices (sensors) form a complete graph. The dependence of reliability on the number of sensors and the threshold value is shown in Figure 1.

Please note that the network reliability in this case does not depend on the number of sinks, while the other parameters have a significant impact. For example, through increasing the number of sensors, *N*, it is possible to achieve an arbitrarily high level of reliability, even with extremely low individual sensor reliability. However, for other topologies, the number of sensors may not have any effect on the considered indicator of network reliability.

Let the WSN topology be a simple chain with one sink at the end. The reliability criterion will be the probability that only the *T* sensors closest to the drain are reliable, as follows:(5)$$R_{S,T}\left(G\right)=R_{\bar{S},T}(G)=p^{T}$$

Adding a second sink at the opposite end of the chain has a fundamental impact on the network’s reliability, with opposite effects on the metrics $R_{S,T}\left(G\right)$ and $R_{\bar{S},T}(G)$. Due to the sink connectivity requirement, the following is calculated:(6)$$R_{S,T}\left(G\right)=p^{N}$$
i.e., the reliability indicator deteriorates $p^{T-N}$ times. At the same time,
(7)$$R_{\bar{S},T}\left(G\right)={\left(T+1\right)\;p}^{T}-Tp^{T+1}$$

Thus, in the absence of a requirement for sink connectivity, the reliability indicator does not depend on the total number of sensors and can be significantly improved by adding a sink, as illustrated in Figure 2.

Next, we illustrate the application of the proposed algorithms to the problem of optimal sink placement in a WSN. We consider a 5 × 5 grid network topology where three sink nodes must be placed to maximize the reliability of the resulting topology. We use $R_{S,T}(G)$ and $R_{\bar{S},T}(G$*)* as reliability measures. In the first scenario, we maximize the probability that at least *T* sensors have access to the sinks, ensuring sink connectivity. In the second scenario, we maximize the probability that at least *T* sensors have access to the sinks without considering sink connectivity.

We use three different values of *T*, 10, 15, and 20. For each *T* value, we consider three sensor reliability values, 0.1, 0.5, and 0.9, assuming the same reliability across all sensors. Initially, a reference solution will be obtained. We solved the problem through an exhaustive search of all sensor placement variants, precisely calculating the reliability for each configuration. Let us consider $R_{S,T}(G)$ as the reliability measure. Distinct solutions based on sensor reliability values were observed when *T* = 15 (Figure 3). Conversely, when *T* is set to 10 or 20, the optimal placements of the sinks remain consistent regardless of the sensor reliability values (Figure 4).

It is noteworthy that for *T* = 20, four non-isomorphic optimal sink placements exist, each achieving the maximum reliability value for its respective topology. These placements involve positioning sinks on the outer boundary of the grid, adhering to the condition that three sinks, denoted as *u*, *v*, and *w*, partition the grid’s outer boundary into three parts.

Let us denote the number of edges in such a part between *i* and *j* (*i*, *j ∈* {*u*, *v*, *w*}) by *d(i*, *j)*. If and only if *d(i*, *j)* is even and 2 ≤ *d(i*, *j)* ≤ 10 for *i ≠ j*, *i*, *j ∈ {u*, *v*, *w}*, then *u*, *v*, *w*, are the optimal nodes for placing the sinks. The total number of all such placements is 32.

In most cases, the results of the proposed heuristic algorithm coincide with the results of the exhaustive search. There are only a few exceptions (Figure 5) for *T* = 15, *p* = 0.1, 0.5, and for *T* = 10, *p* = 0.1, 0.5, when the heuristic algorithm is inferior to an exhaustive search.

During the calculation process, we use different initial solutions. However, for each combination of *T* and *p*, the result obtained does not depend on the initial solution. Almost every case demands two main cycles of the above-mentioned heuristic algorithm and, hence, the reliability analysis of 133 network topologies for various sink placements. For *T* = 15, *p* = 0.9, the heuristic algorithm terminates after three main cycles, which corresponds to the reliability analysis of 199 network topologies. The exhaustive search requires the reliability analysis of 2300 network topologies for various sink placements. Therefore, the heuristic algorithm operates for an appropriate time and provides a reliable enough topology that is either optimal or almost optimal.

Note that any sink placement, which is isomorphic to optimal, is obviously also optimal. For both placements shown in Figure 3, there are four isomorphic placements. The placement shown in Figure 4 on the right has two isomorphic configurations, as well as the placement shown in Figure 5 on the right. The placement shown in Figure 5 on the left has eight isomorphic configurations.

For $R_{\bar{S},T}(G)$ as a reliability measure, we obtained the following results. The different placements depending on the sensor reliability value were obtained for *T* = 10 (Figure 6) and for *T* = 15 (Figure 7). The optimal placements coincide for *p* = 0.9 and for *T* values 10 and 15 (Figure 8). For *T* = 20, the results are the same as in the case of $R_{S,T}(G)$.

When calculating $R_{\bar{S},T}(G)$, the results of the heuristic algorithm coincide with the results of the exhaustive search.

## 8. Conclusions

The integration of diverse WSNs inevitably leads to the formation of a network that comprises multiple sinks. Therefore, this paper addresses the critical issue of WSN reliability through an analysis of the availability of a sufficient number of operational sensors capable of transmitting the collected data to any non-empty subset of sink nodes. We evaluate the reliability index under two scenarios—with and without the requirement for sink connectivity—to ensure proper network functionality. We propose algorithms for calculating the reliability in both scenarios and present a heuristic algorithm for optimal sink placement. Rationalizing the placement of sinks significantly impacts system reliability. In the considered grid, when *p* is high, system reliability approaches 1 regardless of sink placement, making the optimal placement only marginally better than random placement (about 2% on average). However, when *p* is around 0.5, random sink placement can result in a system reliability that is 50% worse than the optimal arrangement. For small values of *p*, optimal sink placement can enhance the reliability by orders of magnitude. In a chain-type topology, as we have shown, the reliability improvement is determined by a threshold value and can become arbitrarily large for any *p*. Despite the NP-hard nature of these problems, the proposed algorithms are applicable to small- and medium-scale WSNs. Moreover, our approach provides a foundation for other WSN reliability measures tailored to specific application domains. For example, the proposed algorithm can be modified to accommodate a limited number of transit nodes in multi-hop communication between sensors and sinks. In future work, we aim to develop methods for estimating topology reliability in larger networks without necessitating exhaustive and computationally intensive calculations of graph reliability. These methods will be based on adapted reliability cumulative updating techniques, Monte Carlo methods, machine learning techniques, and bionic algorithms.

## Figures and Tables

**Figure 1 sensors-24-05468-f001:**
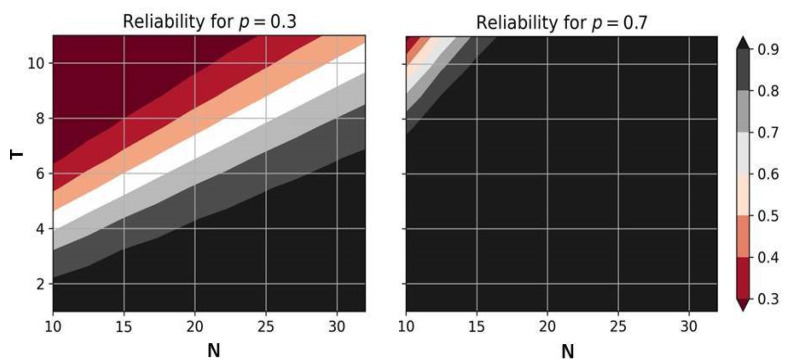
The reliability of star topology and multi-sink topologies with analogous properties.

**Figure 2 sensors-24-05468-f002:**
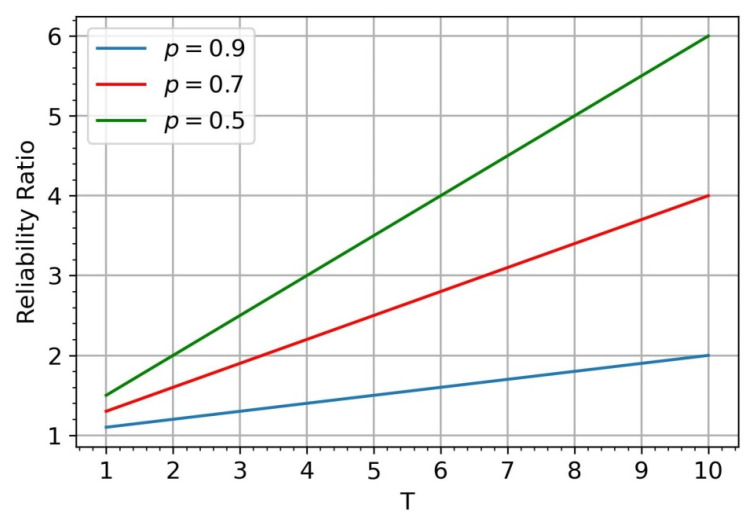
The ratio of reliability indicator $R_{\bar{S},T}\left(G\right)$ in the cases of 2 sinks and 1 sink.

**Figure 3 sensors-24-05468-f003:**
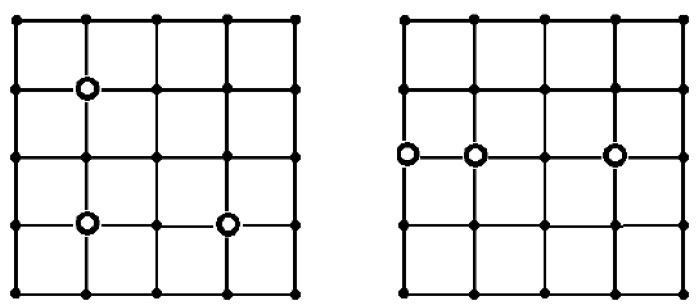
The optimal placement of sinks considering the reliability index $R_{S,T}(G)$ for *T* = 15. The sensor reliability is as follows: *p* = 0.1, 0.5 (**on the left**), and *p* = 0.9 (**on the right**).

**Figure 4 sensors-24-05468-f004:**
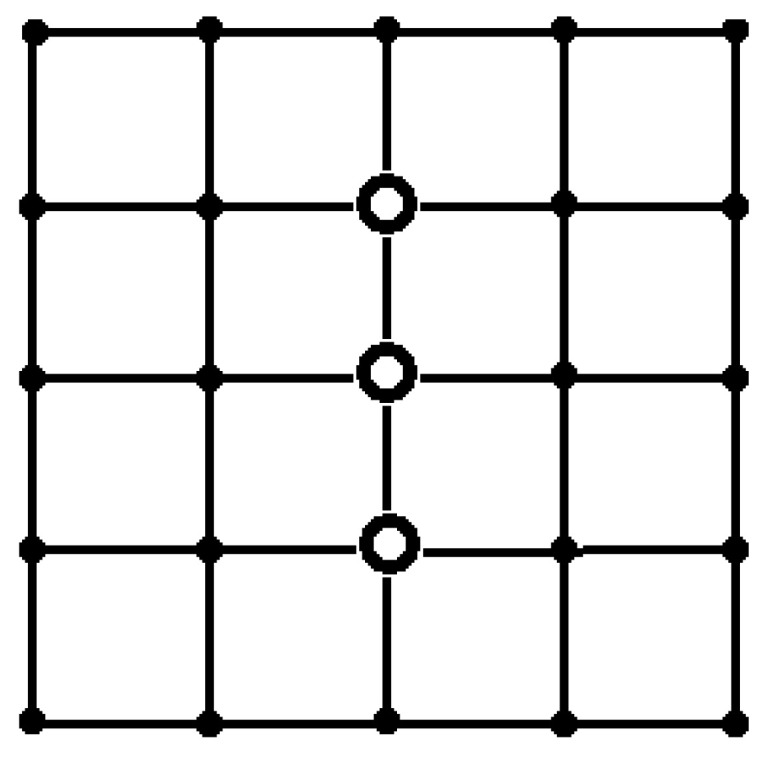
The optimal placement of sinks considering the reliability index $R_{S,T}(G)$ for *T* = 10 (or 20).

**Figure 5 sensors-24-05468-f005:**
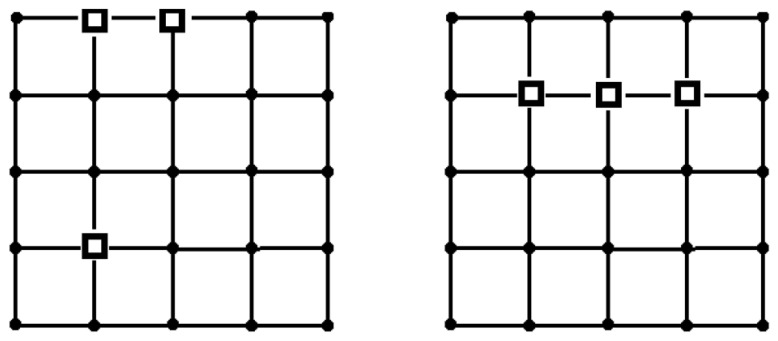
The results of the heuristic algorithm for the placement of sinks at *T* = 15, *p* = 0.1, 0.5 (**on the left**), and *T* = 10, *p* = 0.1, 0.5 (**on the right**).

**Figure 6 sensors-24-05468-f006:**
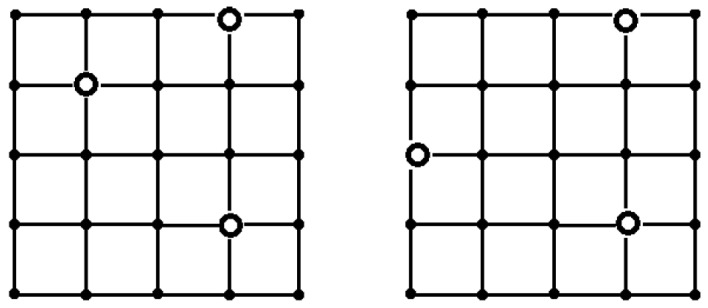
The optimal placement of sinks in term $R_{\bar{S},T}\left(G\right)$ for *T* = 10, *p* = 0.1 (**on the left**), and *p* = 0.5 (**on the right**).

**Figure 7 sensors-24-05468-f007:**
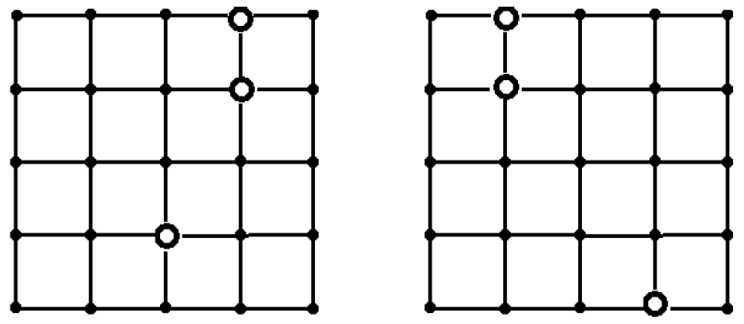
The optimal placement of sinks in term $R_{\bar{S},T}\left(G\right)$ for *T* = 15, *p* = 0.1 (**on the left**), and *p* = 0.5 (**on the right**).

**Figure 8 sensors-24-05468-f008:**
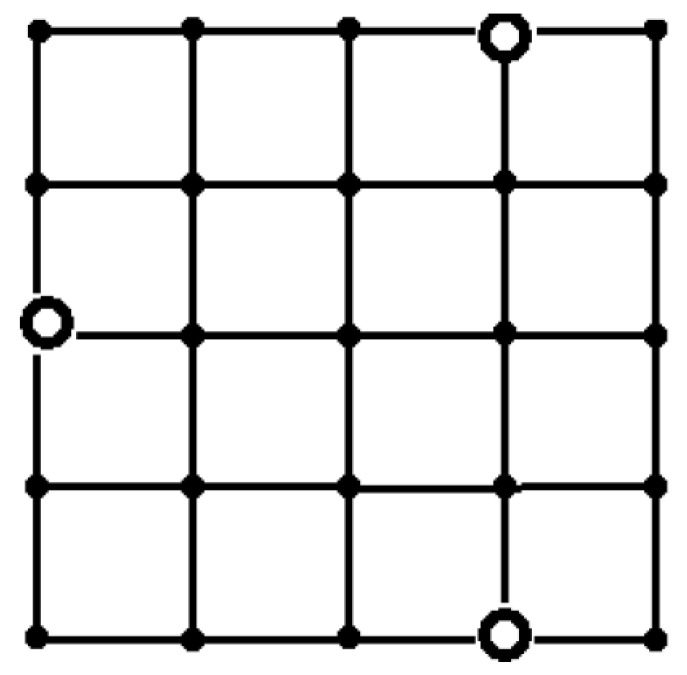
The optimal placement of sinks in term $R_{\bar{S},T}\left(G\right)$ for *T* = 10 or 15, *p* = 0.9.

## Data Availability

Data are contained within the article.

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
