# Peer review of "On the Reliability of Wireless Sensor Networks with Multiple Sinks"

_sensors, 2024, doi:10.3390/s24175468_

Round 1

Reviewer 1 Report

Comments and Suggestions for Authors

This paper considers a wireless sensor network consisting of sensors and sinks and studies the reliability of the network defined as the probability that sinks can collect data from a given number of sensors. The paper proposes an algorithm to find this probability given the probabilities of each sensor's outage. Then an algorithm for optimal sink placement is proposed.

The paper is quite good, I have small suggestions to improve it:

1. In Section 3 it is unclear, how the edges between the vertices are generated. We have p_v - probability of an operational vertex, but the edges are unclear. Do the vertices have some coordinates which determine their neighbor relation? If so, how are they distributed in space? Please provide more details in Section 3.

2. "If, in the branch of contraction, the number of sensors attached to the sink reaches T, then we obtain the final graph." — do the authors mean the number of sensors attached to each sink? Or does the algorithm consider only one sink?

Author Response

First, we want to express our thanks to the reviewer for providing valuable comments that help improve paper quality. The paper has been extensively revised according to the reviewers’ comments. Below the authors’ responses are given for individual comments mentioned by the reviewer.

Comment 1

Reviewer’s

Comment:

The paper is quite good, I have small suggestions to improve it: In Section 3 it is unclear, how the edges between the vertices are generated. We have p_v - probability of an operational vertex, but the edges are unclear. Do the vertices have some coordinates which determine their neighbor relation? If so, how are they distributed in space? Please provide more details in Section 3.

Authors’ Reply:

The edges are assumed to be perfectly reliable. If the distance between two sensors does not exceed the transmission range of the sensors, an edge is created between the corresponding vertices in the graph; otherwise, no edge is formed. At the short distances typical in WSNs, Automatic Repeat Request (ARQ) or Forward Error Correction (FEC) mechanisms can ensure near-perfect channel reliability. To reflect the reviewer's comment, we added the requested details to the paper.

Comment 2

Reviewer’s

Comment:

"If, in the branch of contraction, the number of sensors attached to the sink reaches T, then we obtain the final graph." — do the authors mean the number of sensors attached to each sink? Or does the algorithm consider only one sink?

Authors’ Reply:

To reflect the reviewer’s comment, we clarified the criterion.

Reviewer 2 Report

Comments and Suggestions for Authors

The introduction should explore more in term of current state of the metric, which clarifying the concrete and details examples in practical situation in regard of dedicated and complex concept that was presented in the paper.

The explanation in the related works should be compared and put it in the table to show the difference between them especially in understanding the advancement and the lack. It can help the reader to validate the effectiveness of the proposed algorithms by emphasizing the metric. In understanding the computational feasibility also critical for practical application, so the exploration further should be provided to address the issue of scalability, performance, accuracy and efficiency.

Use proper citations by adding the name of the authors rather just the order of citation like: In [17], the authors investigated the reliability of a four-fiber bidirectional line-switched ring; it should be In Cholda et. al. [17], they investigated the reliability of a four-fiber bidirectional line-switched ring.

Testing the algorithms on real or simulated datasets would be beneficial to show its effectiveness in various scenarios and provide more discussion in interpreting the findings and understanding the impact. The authors also can provide more detailed descriptions of the algorithms with flowcharts may in assisting the code.

Various risk should be considered as well especially in regard to overfitting to certain datasets or scenarios that lead to bias. The network configuration should be presented and the mechanism that address the way for generalizability as well. Certain scenarios should be explored more like dynamic changes in the networks especially the motivation for doing them. At some extent, diverse and different type of sensor nodes and capabilities can present unexpected network faults. Parameter sensitivity also can be conducted to understand how changes occurred and influence the reliability metrics.

The paper also mentioned about certain failure rates or pattern for particular sensors, thus it will be useful to examine the alignment with real-world scenarios that address various factors such as interference or mobility. Meanwhile, the energy and data quality also important, which the proposed metrics can integrate and discuss this concern in WSNs.

Comments on the Quality of English Language

The sentences with multiple clauses and technical jargon can be simplified. Also, the ideas and points that are repeated can be consolidated such as the word of network reliability that often-occurred numerous times.

Author Response

First, we want to express our thanks to the reviewer for providing valuable comments that help improve paper quality. The paper has been extensively revised according to the reviewers’ comments. Below the authors’ responses are given for individual comments mentioned by the reviewer.

Comment 1

Reviewer’s

Comment:

The introduction should explore more in term of current state of the metric, which clarifying the concrete and details examples in practical situation in regard of dedicated and complex concept that was presented in the paper.

Authors’ Reply:

To reflect the reviewer's comment, we add the requested details to the Introduction.

Comment 2

Reviewer’s

Comment:

The explanation in the related works should be compared and put it in the table to show the difference between them especially in understanding the advancement and the lack. It can help the reader to validate the effectiveness of the proposed algorithms by emphasizing the metric. In understanding the computational feasibility also critical for practical application, so the exploration further should be provided to address the issue of scalability, performance, accuracy and efficiency.

Authors’ Reply:

To the best of our knowledge, the problem statement presented in this paper is original and the algorithms we present have no known counterparts. To reflect the reviewer’s comment, we mentioned it in the paper.

Comment 3

Reviewer’s

Comment:

Use proper citations by adding the name of the authors rather just the order of citation like: In [17], the authors investigated the reliability of a four-fiber bidirectional line-switched ring; it should be In Cholda et. al. [17], they investigated the reliability of a four-fiber bidirectional line-switched ring.

Authors’ Reply:

We agree that this citation style may look more exquisite, but we are guided by the result of the MDPI English language correction service with the style correction option enabled. Also, we have used this style in our previous papers published in MDPI Sensors, and we need to be consistent. Please let us know if the citation style requirements have changed.

Comment 4

Reviewer’s

Comment:

Testing the algorithms on real or simulated datasets would be beneficial to show its effectiveness in various scenarios and provide more discussion in interpreting the findings and understanding the impact. The authors also can provide more detailed descriptions of the algorithms with flowcharts may in assisting the code.

Various risk should be considered as well especially in regard to overfitting to certain datasets or scenarios that lead to bias. The network configuration should be presented and the mechanism that address the way for generalizability as well. Certain scenarios should be explored more like dynamic changes in the networks especially the motivation for doing them. At some extent, diverse and different type of sensor nodes and capabilities can present unexpected network faults. Parameter sensitivity also can be conducted to understand how changes occurred and influence the reliability metrics.

Authors’ Reply:

In this paper, we present exact graph-theoretic algorithms, so we do not use datasets and do not face the problem of overfitting. We plan to investigate the possibility of using ML/DL methods to estimate the reliability of random graphs and, hence, we will meet these issues. However this is our future work. To reflect the reviewer's comment, we have removed ambiguities in the pseudocode, thereby making it easier to transform into working code. Also, we included ML-based technique to future work.

Comment 5

Reviewer’s

Comment:

The paper also mentioned about certain failure rates or pattern for particular sensors, thus it will be useful to examine the alignment with real-world scenarios that address various factors such as interference or mobility. Meanwhile, the energy and data quality also important, which the proposed metrics can integrate and discuss this concern in WSNs.

Authors’ Reply:

In this paper, we pay special attention to the structural reliability of WSNs. However, the presented results can also be used to calculate the functional reliability indicators if the values ​​of the probabilities of the existence of random graph elements are calculated using auxiliary models responsible for the mobility of the sensor, energy consumption, etc. The corresponding technique has been presented in the paper [27]. To reflect the reviewer's comment, we have mentioned this technique.

Comment 6

Reviewer’s

Comment:

The sentences with multiple clauses and technical jargon can be simplified. Also, the ideas and points that are repeated can be consolidated such as the word of network reliability that often-occurred numerous times.

Authors’ Reply:

To improve the quality of English, we use the MDPI English correction service with the additional option to correct style.

Reviewer 3 Report

Comments and Suggestions for Authors

The article is devoted to the current topic of assessing and increasing the wireless sensor networks reliability with multiple receivers. The algorithms proposed in the work make it possible to estimate the WSN functioning probability if T of N sensors remain operational with or without the need to connect to sink(s).

After analyzing the paper, I conclude that the following things need to be corrected:

1. In section 3, an elementary event is denoted by the symbol , which is then not used anywhere.

2. In Section 4, the breadth-first search algorithm computational complexity is denoted as O(|M|), and the variable M is not described anywhere. It is known that the breadth-first search algorithm computational complexity in the authors’ notation is defined as O(|V|+|E|). Why was the variable M introduced, or does it express the specifics of the algorithm developed by the authors?

3. Line 17 of Algorithm 1 uses the constant S, which, according to the description, is initialized as zero. Moreover, it seems to me that it should clearly have a non-zero value. By the way, in the second algorithm this letter denotes set of nodes that are not sinks.

4. Line 9 of Algorithm 2 uses the undocumented set K.

5. Further, when modeling the optimization algorithm on a 5x5 grid, in the expression for reliability, the symbol 𝔖 is for some reason replaced by K. Is this a typo, or is this the undocumented set K?

6. I would also like to obtain numerical estimates of the increase in the WSN reliability through the use of an algorithm for optimizing the sinks' location for the considered examples, which, in my opinion, should be added to the conclusion.

Author Response

First, we want to express our thanks to the reviewer for providing valuable comments that help improve paper quality. The paper has been extensively revised according to the reviewers’ comments. Below the authors’ responses are given for individual comments mentioned by the reviewer.

Comments of the Reviewer 3 and the Authors’ response

Comment 1

Reviewer’s

Comment:

In section 3, an elementary event is denoted by the symbol, which is then not used anywhere.

Authors’ Reply:

We are grateful to the reviewer for his careful proofreading. To reflect the reviewer’s comment, we have removed redundant information in Section 3.

Comment 2

Reviewer’s

Comment:

In Section 4, the breadth-first search algorithm computational complexity is denoted as O(|M|), and the variable M is not described anywhere. It is known that the breadth-first search algorithm computational complexity in the authors’ notation is defined as O(|V|+|E|). Why was the variable M introduced, or does it express the specifics of the algorithm developed by the authors?

Authors’ Reply:

We agree that in general the computational complexity of BFS is O(|V|+|E|). In this particular case, |V| is less than or equal to |E|+1, therefore,  the complexity can be defined as O(|E|). To reflect the reviewer's comment, we fixed this point.

Comment 3

Reviewer’s

Comment:

Line 17 of Algorithm 1 uses the constant S, which, according to the description, is initialized as zero. Moreover, it seems to me that it should clearly have a non-zero value. By the way, in the second algorithm this letter denotes set of nodes that are not sinks.

Authors’ Reply:

To reflect the reviewer's comment, we fixed this point.

Comment 4

Reviewer’s

Comment:

Line 9 of Algorithm 2 uses the undocumented set K.

Authors’ Reply:

We are grateful to the reviewer for his careful proofreading. To reflect the reviewer's comment, we have corrected this typo.

Comment 5

Reviewer’s

Comment:

Further, when modeling the optimization algorithm on a 5x5 grid, in the expression for reliability, the symbol ? is for some reason replaced by K. Is this a typo, or is this the undocumented set K?

Authors’ Reply:

We again appreciate the reviewer’s careful observations. To reflect the reviewer's comment, we have corrected this typo.

Comment 6

Reviewer’s

Comment:

I would also like to obtain numerical estimates of the increase in the WSN reliability through the use of an algorithm for optimizing the sinks' location for the considered examples, which, in my opinion, should be added to the conclusion.

Authors’ Reply:

To reflect the reviewer’s comment, we added this information to the conclusion.
